# Statins—Their Role in Bone Tissue Metabolism and Local Applications with Different Carriers

**DOI:** 10.3390/ijms25042378

**Published:** 2024-02-17

**Authors:** Marcin Mateusz Granat, Joanna Eifler-Zydel, Joanna Kolmas

**Affiliations:** 1Department of Clinical and Experimental Pharmacology, Faculty of Medicine, Medical University of Warsaw, Banacha 1 Str., 02-097 Warsaw, Poland; marcin.granat@wum.edu.pl; 2Department of Pharmaceutical Chemistry and Biomaterials, Faculty of Pharmacy, Medical University of Warsaw, Banacha 1 Str., 02-097 Warsaw, Poland; joanna.eifler@gmail.com

**Keywords:** statins, simvastatin, local application, bone tissue metabolism, statin carriers

## Abstract

Statins, widely prescribed for lipid disorders, primarily target 3-hydroxy-3-methylglutaryl-coenzyme A (HMG-CoA) reductase competitively and reversibly, resulting in reduced low-density lipoprotein cholesterol (LDL-C). This mechanism proves effective in lowering the risk of lipid-related diseases such as ischemic cerebrovascular and coronary artery diseases. Beyond their established use, statins are under scrutiny for potential applications in treating bone diseases. The focus of research centers mainly on simvastatin, a lipophilic statin demonstrating efficacy in preventing osteoporosis and aiding in fracture and bone defect healing. Notably, these effects manifest at elevated doses (20 mg/kg/day) of statins, posing challenges for systematic administration due to their limited bone affinity. Current investigations explore intraosseous statin delivery facilitated by specialized carriers. This paper outlines various carrier types, characterizing their structures and underscoring various statins’ potential as local treatments for bone diseases.

## 1. Introduction

Statins, originally derived from mold fungi, constitute a class of medications widely employed in treating lipid disorders, which stand as a primary risk factor for atherosclerosis and related conditions like coronary artery disease, ischemic cerebrovascular disease, and peripheral vascular diseases [1,2,3]. These compounds function as competitive, reversible inhibitors of the enzyme 3-hydroxy-3-methylglutaryl-coenzyme A (HMG-CoA) reductase, a key catalyst in the early stages of hepatic cholesterol biosynthesis. Approximately two thirds of the body’s cholesterol is synthesized de novo in the liver, with the transformation mediated by HMG-CoA reductase serving as the rate-limiting step in this process [4].

Inhibiting this enzyme’s activity diminishes cellular cholesterol levels, prompting the upregulation of low-density lipoprotein (LDL) receptors on the cell surface and enhancing the cellular uptake of LDL from the bloodstream. The consequential reduction in plasma LDL stands as the primary therapeutic effect of statins [4,5]. Additionally, these drugs exhibit the capability to modestly decrease triglyceride levels, elevate high-density lipoprotein (HDL) concentration, and reduce lipid susceptibility to oxidation [6]. Moreover, their cardioprotective effects may extend beyond lipid profile modulation, possibly involving increased endothelial nitric oxide synthesis, leading to improved vascular endothelial function [7].

Statins might mitigate cardiovascular event risk by stabilizing atherosclerotic plaques, inhibiting metalloproteinase secretion by macrophages, and impeding extracellular matrix breakdown, thus fortifying plaque fibrous membranes [8,9]. This drug class also influences the coagulation system, diminishing platelet aggregation and promoting fibrinolysis [6,10]. Statins may further modulate arterial wall cell numbers by restricting smooth muscle cell proliferation and enhancing apoptosis [11]. Notably, they exhibit antioxidant, immunomodulatory, and anti-inflammatory effects, addressing the emerging emphasis on chronic inflammation’s role in atherosclerosis development [12,13]. Their pleiotropic effects and generally favorable patient tolerance make statins a cornerstone in cardiovascular disease treatment (see Figure 1).

Commonly utilized statins include mevastatin, lovastatin, and pravastatin (natural statins); simvastatin (a semi-synthetic statin); and fluvastatin, rosuvastatin, pitavastatin, cerivastatin, and atorvastatin (synthetic statins) [14,15]. Their chemical formulas are depicted in Figure 2. Notably, variations in their chemical structures impact lipophilicity, hydrophilicity, and subsequent absorption, distribution, metabolism, and excretion [14,16].

The molecule of each statin consists of three main parts: an HMG-CoA analogue, a complex ring structure responsible for binding the statin to HMG-CoA reductase, and a side chain structure determining solubility. Statins vary greatly in solubility due to the presence/absence of polar moieties in their predominantly hydrophobic backbones. Lipophilic statins can easily penetrate deeper into cell membranes and enter cells through passive diffusion, and are therefore widely distributed in various tissues. They are metabolized by cytochrome P450 (CYP) enzymes upon binding to the membrane. In turn, hydrophilic statins remain bound to the polar surface of the membrane and require the transport of proteins into the cell, show greater hepatoselectivity and lower potential for uptake by peripheral cells than lipophilic statins, and are mostly eliminated unmetabolized.

Upon oral administration, statins undergo substantial hepatic first-pass effects, limiting their systemic bioavailability to 5–30% of the administered dose due to high hepatic uptake. In plasma, nearly 95% of statins and their metabolites are protein-bound [14,16]. Some, like simvastatin, are administered as inactive prodrugs requiring hydrolysis catalyzed by esterases and peroxidases for therapeutic effect. Despite generally good tolerability, serious adverse effects, especially at higher doses, may occur. Serving as substrates for CYP3A4 (cytochrome P450 family 3 subfamily A member 4) isoenzyme and OATP-2 (organic anion transporting polypeptide-2), statins can engage in clinically significant interactions with other drugs affecting these proteins.

Common adverse effects of statin therapy include elevated liver transaminases, with hepatotoxicity and nephrotoxicity as potential risks. Some rare but severe adverse effects are myopathy and rhabdomyolysis, with the risk directly proportional to statin concentration in plasma [17,18].

Currently, research explores statin use beyond cardiac diseases related to lipid disorders, including potential applications in bone diseases, injuries, and tissue defects [19,20].

Our study focuses on investigating the effects of statins on bone tissue metabolism, considering different carriers for drug application.

It should be noted that statins, like most drugs, are typically administered orally. However, the oral administration of statins presents drawbacks, such as first-pass metabolism in the liver and degradation in the gastrointestinal tract, resulting in limited bioavailability. Another concern is the occurrence of adverse effects such as myopathy, kidney and liver damage, and rhabdomyolysis. Hence, alternative routes of statin administration are under investigation, particularly for potential use in bone diseases where prolonged action directly within the bone tissue is required.

We conducted a comprehensive search on databases such as PubMed, Embase, IEEE Explorer, and Google Scholar, accessing them on 27 December 2023, due to their leading role in scientific research and English-language content. The search phrases included “statin bone tissue”, “statin bone metabolism”, “statin bone application”, “statin bone”, “statin osteoporosis”, and “statin biomaterials”. The publications considered spanned from 1999 to 2023, complemented by insights from pharmacology and pharmaceutical chemistry books, contributing approximately 15% of the overall information.

## 2. Statins—Their Role in Bone Tissue Metabolism

As early as the late 20th century, observations indicated that statins can stimulate osteogenesis and offer therapeutic benefits to patients with osteoporosis [21,22,23,24,25]. Subsequent research has unveiled their potential as robust modulators of bone healing and the inflammation process, demonstrating their capacity to stimulate osteoinduction and angiogenesis—key phenomena for bone reconstruction [26,27]. They also exhibit the ability to limit the loss of bone tissue by inhibiting both osteoclastogenesis and the apoptosis of osteoblasts [28,29,30]. Therefore, the beneficial effect of statins on bone tissue metabolism results from their action through various mechanisms, as confirmed in numerous animal models of osteoporosis and mechanical bone damage [31,32,33,34,35]. Lipophilic statins, particularly simvastatin, lovastatin, and atorvastatin, appear to be the most effective in preventing osteoporosis and supporting the healing of bone fractures and defects [22,23,36].

Several mechanisms have been proposed to explain the anabolic effect of statins on bone tissue (see Figure 3). Firstly, these drugs can stimulate the formation of new tissue by increasing the expression of a gene for bone morphogenetic proteins (BMPs), specific growth factors regulating osteoblast and chondroblast proliferation and differentiation [21,26,37,38]. Statins increase BMP-2 expression by modulating small GTP-binding proteins, particularly the Ras and Rho proteins [37]. It is important to note that only lipophilic statins can activate the promoter of this gene. Another suggested mechanism related to the primary target point of statins involves blocking HMG-CoA reductase, inhibiting the conversion of HMG-CoA to mevalonate, and indirectly limiting the biosynthesis of its derivatives, such as farnesyl pyrophosphate or geranylgeranyl pyrophosphate [22,36,37,38,39]. These derivatives adversely affect bone tissue metabolism. For example, farnesyl diphosphate can activate the glucocorticoid receptor, potentially increasing bone resorption and reducing osteoblast proliferation. Geranylgeranyl pyrophosphate likely also negatively affects osteogenesis [21,22]. By affecting the TGFβ/Smad3 pathway, statins protect osteoblasts against apoptosis. TGFβ (transforming growth factor beta) activates Smad3, promoting the formation of new bone tissue by stimulating the synthesis of extracellular matrix proteins, intensifying mineralization, and increasing alkaline phosphatase activity—a marker of early osteoblast differentiation (see Figure 3) [28,30].

Studies have shown that bone marrow stem cells in the presence of statins exhibit increased activity of this protein, with the drug released from the carrier stimulating the mineralization of the extracellular matrix [40,41,42,43]. The loss of Smad3 results in the apoptosis of osteoblasts and osteocytes, leading to decreased bone mass. By enhancing its expression in these cells, statins may contribute to their protection. These drugs can also modulate osteoclastogenesis by inhibiting it through the OPG/RANKL/RANK pathway [40]. RANKL (receptor activator of NF-κB ligand), RANK (receptor activator of NF-κB), and osteoprotegerin (OPG) play key roles in this process. OPG reduces the proliferation of osteoclasts, while RANK, belonging to the tumor necrosis factor receptor (TNFR) superfamily, enhances osteoclastogenesis. RANKL, being a membrane protein on the surface of osteoblasts, acts as a key factor for osteoclast function by blocking their apoptosis [26]. Mice lacking RANKL or RANK developed significant osteoporosis due to the complete absence of osteoclasts and the consequent lack of bone resorption and remodeling. In turn, OPG-deficient mice showed osteoporosis resulting from an increased number of osteoclasts. The balance of this signaling pathway is crucial for proper, balanced bone remodeling, and statins can modulate it.

In studies on a mouse osteocyte cell line, these drugs have been shown to increase the expression of the OPG gene and decrease the expression of the RANKL gene [22,24]. It has also been proven that the number of osteoclasts decreases with increasing statin concentration. Studies also suggest that statins may promote neovascularization because they increase gene expression for vascular endothelial growth factor (VEGF) and fibroblast growth factor 2 (FGF-2). The processes of osteogenesis and angiogenesis are closely related, and coordinated vascularization is of key importance in the process of bone tissue reconstruction. It not only provides oxygen and nutrients to the developing tissue, but also allows the transport of mesenchymal stem cells from the bone marrow, further supporting bone regeneration [44,45,46,47,48,49].

The ability of statins to simultaneously stimulate these two processes makes them even more promising potential drugs in the therapy of bone damage and defects [50,51]. Unfortunately, systemically administered statins have a low affinity for bone tissue, making the systemic use of these drugs in attempts to treat bone diseases potentially inconclusive as to their effectiveness [51,52]. Additionally, as mentioned previously, most statins undergo a significant first-pass effect, further limiting their bioavailability in the bones. Therefore, orally administered statins in clinical doses required for lipid-lowering therapy are not able to exert a strong positive effect on the anabolism and catabolism of bone tissue [51,52,53]. Achieving these effects would require administering drugs at much higher doses, significantly increasing the risk of systemic adverse effects. Therefore, the topical application of statins directly to the bone tissue may be an alternative option that allows for bypassing the hepatic metabolism of these drugs, increasing their bioavailability, and reducing the dose. Other resulting benefits include reducing the risk of systemic adverse effects, as well as eliminating the need for repeated administration of the drug.

## 3. Statin Local Delivery Methods and Carriers

The local administration of statins in the case of bone tissue disorders seems justified for several reasons. Firstly, bone tissue is highly vascularized, and systemic treatment requires the use of large doses of therapeutic substances to achieve an adequate concentration at the target site [54,55]. Therefore, the direct administration of statins to bone tissue allows for a reduction in dosage compared to systemic administration [56]. Additionally, the carrier for intraosseous drug delivery can provide a matrix for the infiltration of mesenchymal cells and often serves as a filling material for bone defects [57,58]. It is essential to emphasize that the optimal carrier should exhibit appropriate degradation rates to ensure the proper growth of newly formed bone tissue while preventing the formation of fibrous tissue and fibrous encapsulation of the carrier. The local administration of statins can effectively reduce the risk of adverse effects, primarily myopathy characterized by muscle pain and weakness, the elevation of creatine kinase (CK) levels in the serum, and, in extreme cases, rhabdomyolysis, which is a life-threatening condition [57,58].

It is worth noting that our study did not focus on the specific doses of statins used in individual research. The results regarding statin dosage in the treatment of bone tissue diseases remain inconclusive. Depending on the dose, these drugs can exhibit either positive or negative effects on bone health. For instance, high doses of simvastatin (20 mg/kg bw/day) have been shown to stimulate new bone tissue formation, while low doses (1 mg/kg bw/day) may inhibit reconstruction and increase bone resorption [59]. Despite promising results, the practical applications of statins in treating bone diseases require further research, including optimizing dosage and exploring effective methods of application.

Table 1 presents the literature review results regarding carriers for statin delivery into the bone tissue. The data indicate that statin delivery systems used mainly simvastatin; in a few cases, lovastatin [60,61,62]; pitavastatin [63]; rosuvastatin [64,65,66]; and fluvastatin [67,68,69,70,71].

Carriers were equipped with statins in various ways. The simplest method involves impregnating bone scaffolds, but this has the drawback of rapid drug dissolution in the bloodstream, leading to reduced bioavailability. Alternatively, spray techniques and ultrasonic dispersion can coat statins onto bone scaffolds, allowing for only low-dose application. Other approaches involve micro- and nanospheres, chemical compound combinations, polymerization, and blending with biomaterials, each with its advantages and limitations.

Based on their chemical composition, carriers can be categorized into those made of inorganic materials, natural polymers, synthetic polymers, and polymer/inorganic material composites. The choice of carrier materials is dictated by the structure and properties of bone tissue, composed of an organic matrix (mainly collagen type I fibers) and an inorganic phase, primarily biological apatite, a nanocrystalline calcium phosphate. Inorganic carriers offer high compatibility, bioactivity, and bioresorbability, but may lack sufficient mechanical strength, especially when serving as bone defect fillers. On the other hand, polymers, despite high biocompatibility, may lack the necessary hardness. Therefore, composite carriers seem to be the optimal solution, providing adequate mechanical properties, biodegradation, and controlled statin release at the target site.

In the case of inorganic carriers, calcium phosphates are primarily utilized due to their similarity to biological apatite. Among the most commonly used calcium phosphates, hydroxyapatite and crystalline calcium orthophosphates (in both low- and high-temperature forms; respectively, β and α) can be mentioned, with α-TCP being a compound that is much more soluble than β-TCP and hydroxyapatite. α-TCP has been employed in the form of bone cement, a material that solidifies in reaction with water to form a hard filling for bone defects. β-TCP has also been used in the form of cement, as well as a biphasic material with hydroxyapatite, thereby increasing the material’s bioresorbability and bioactivity. In the study [72], unconventional hydroxyapatite in the form of nanofibers was used, while in [73], together with simvastatin, it served as a coating for a titanium implant. In this case, hydroxyapatite was used as a factor for improving osteointegration and as a carrier for delivering a statin. The studies [74,75] focus on the use of a scaffold of calcium sulfate loaded with simvastatin. Research conducted using inorganic materials was carried out in both in vitro conditions (primarily on BMSCs cells) and in vivo animal models (rats, mice, and rabbits). In all the presented studies, the simultaneous administration of the inorganic material with statin resulted in the increased formation of new bone tissue, increased density of the created tissue, and improvement in bone tissue regeneration, which were observed between the 3rd and 12th week of the experiment.

Significantly more research has focused on the utilization of both natural and synthetic polymers, as well as composite materials, as potential carriers for statins. This emphasis likely stems from the inherent drawbacks of inorganic materials (primarily calcium phosphates), notably, their brittleness and, thus, insufficient mechanical strength, which render them unsuitable for areas subjected to high stresses. Polymers are characterized by high flexibility, whereas calcium phosphate/polymer composites combine the hardness of inorganic materials with the elasticity of polymers, ensuring excellent mechanical properties akin to those of bone tissue, which is also a natural composite.

Among the natural polymers utilized for statin carriers, chitosan has been predominant, with collagen or prepared atelocollagen being alternative materials. Gelatin has also been employed in hydrogel form. Concerning synthetic polymers, a wide array have been utilized, including both biodegradable (e.g., PLA, PLLA, PCL, PLGA) and non-biodegradable (PUR, PEEK) polymers. The composite materials described in the available literature consist of hydroxyapatite or beta-TCP combined with polymers such as poly(e-caprolactone), poly(glycerol sebacate), gelatin-nanofibrillar cellulose, PCL, and PLGA. Polymer and composite materials are prepared in various forms, including porous scaffolds, microspheres, nanomicelles, fibers, nanoparticles, and coatings on titanium surfaces. The available literature predominantly comprises in vivo studies on rats and rabbits, which consistently indicate increased mineralization, bone tissue formation, and improvements in bone parameters in each case.

A significant aspect investigated under laboratory conditions is the release of statins from previously obtained carriers. Various release profiles have been observed depending on the chemical composition of the carrier, its form, and its preparation method. Generally, the constructed carriers are designed to release statins gradually over an extended period (from 50–90% within 7 days) [76,77,78]. In the study by [78], a typical microporous structure of calcium phosphate foam was employed, which proved to be a suitable carrier for pitavastatin: the “burst release” effect was relatively minimal, followed by the gradual release of the statin over 72 h. Various approaches have been applied to prolong the release time. For instance, delayed release was achieved in carriers derived from biomimetic beta-TCP by coating it with an additional layer of apatite [63]. This resulted in a matrix that released 20% less simvastatin over 7 days. In the study by [79], significantly prolonged release of simvastatin was obtained from PCL microspheres contained within a collagen coating covering PET polymer (constructing artificial ligaments). Over 14 days, slightly over 50% of the drug was released, and approximately 75% within nearly 40 days.

**Table 1 ijms-25-02378-t001:** The effects of various statins combined with different types of carriers on bone.

Statin (Dose)	Carrier	Model of Study/Duration of Treatment	Findings	Reference
Inorganic materials				
Simvastatin (0.25 and 0.5 mg)	α-TCP (α-tricalcium phosphate)	In vivo/69 healthy Wistar adult rats; 8 weeks	Bone regeneration in rat calvarial defects was noticed	[80]
Simvastatin (0.1 mg)	α-TCP (α-tricalcium phosphate)	In vivo/72 healthy Wistar rats; 8 weeks	Stimulation of bone regeneration occurred	[81]
Simvastatin (6% concentration)	Apatite cements	In vivo/18 ovariectomized rats; 3 weeks	Bone mineral density increased	[82]
Simvastatin (4 mg/mL)	β-TCP (β-tricalcium phosphate)	In vitro/drug release in simulation body fluid solution; 7 days	Controlled release of the drug with a reduction of approximately 25% compared to control samples was observed	[76]
Simvastatin (0.1 mg)	β-TCP (β-tricalcium phosphate)	In vivo/72 healthy Wistar rats; 8 weeks	Stimulation of bone regeneration occurred	[81]
Simvastatin (0.1, 0.9, and 1.7 mg)	β-TCP (β-tricalcium phosphate)	In vivo/162 healthy male Sprague Dawley rats; 6 weeks	Decreased mineral apposition was observed, and after 26 weeks, increased fibrous area fraction, β-TCP area fraction, and particle size and number were noticed	[83]
Simvastatin (0.1 mg)	Calcium phosphate	In vivo/15 healthy female Wistar rats; 8 weeks	Bone-like tissue was formed	[84]
Simvastatin (1, 5, and 10% concentrations)	Calcium phosphate cement (an equimolar mixture of tetracalcium phosphate and dicalcium phosphate anhydrous)	In vivo/40 healthy New Zealand white rabbits; 4 weeks	New bone formation was observed	[85]
Simvastatin (0.5 and 0.25 mg/g cement)	Calcium phosphate cement	In vitro/Saos-2 cells; 7 days	Promotion of bone formation was noticed	[86]
Simvastatin (0.1, 0.25 and 0.5 mg/g cement)	Calcium phosphate cement (β-tricalcium phosphate and monocalcium phosphate anhydrous in molar ratio of 1:1)	In vitro/bone marrow macrophages isolated from mice; 12 days	Inhibition of osteoclastic differentiation was observed	[87]
Simvastatin (0.5 mg)	Calcium sulphate	In vivo/18 healthy New Zealand white rabbits; 8 weeks	An area of newly formed bone was noticed	[74]
Simvastatin (1 mg)	Calcium sulphate	In vivo/45 healthy male Wistar rats; 8 weeks	Stimulation of bone regeneration was observed	[75]
Simvastatin (0.125 mg)	Hydroxyapatite	In vivo/12 healthy New Zealand white rabbits; 8 weeks	Increased bone volume was noticed	[72]
Simvastatin (0.45 mg)	Hydroxyapatite	In vivo/20 adult Japanese white rabbits; 8 weeks	New bone formation was observed	[73]
Simvastatin (0.1 mg)	Hydroxyapatite	In vivo/72 healthy Wistar rats; 8 weeks	Stimulation of bone regeneration occurred	[88]
Simvastatin (10 mM)	Hydroxyapatite-coated titanium	In vitro/bone mesenchymal stem cells (BMSCs); 14 days In vivo/48 adult male Sprague Dawley rats; 6 weeks	In vitro: Enhanced osteogenesis and osteointegration occurred In vivo: Maximum forces of the Sim-Low and Sim-High groups were significantly higher than those of the Control and HA groups	[88]
Simvastatin (0.01 and 0.001 g/L)	Mesoporous titania thin films	In vitro/MC3T3-E1 pre-osteoblasts cells; 21 days	Incubation the formation of a complex network of pre-collagen filaments was observed	[89]
Simvastatin (10 mM)	Nanohydroxyapatite	In vivo/36 ovariectomized Sprague Dawley rats; 12 weeks	New bone formation around implant surfaces was noticed	[90]
Simvastatin (5 mg/kg)	Titanium implants	In vivo/54 ovariectomized Sprague Dawley rats; 84 days	The bone healing process was observed	[91]
Simvastatin (50 μg/implant)	Titanium Kirschner wires coated with PDLLA (poly(D,L-lactide)) and PUR (polyurethane)	In vivo/200 female Sprague Dawley rats; 6 weeks	Improved fracture healing was present	[92]
Natural and synthetic polymers and composites				
Simvastatin (5, 10, and 20 mg/15 g solutions)	3D—PGHS (as-fabricated 3D fibrous scaffolds of poly (ε-caprolactone) poly (glycerol-sebacate) hydroxyapatite nanoparticles)	In vitro/human mesenchymal stem cells (hMSCs) and human umbilical vein endothelial cells (HUVECs); 7 days	Osteogenic differentiation and migration as well as tube formation occurred	[93]
Simvastatin (10 mg/mL)	ALN-CD (alendronate—β-cyclodextrin) conjugate	In vivo/44 healthy female Sprague Dawley rats; 4 weeks	The study stated that ALN-CD conjugates not only act as tissue-specific carriers but preserve new bone formation	[94]
Simvastatin (2.5 mg/mL dissolved in 0.2 mL water)	ACS (atelocollagen sponge)	In vivo/20 adult male Japanese white rabbits; 12 weeks	New bone formation was observed	[95]
Pitavastatin (0.1 μM)	β-cyclodextrin-grafted chitosan and gelatin	In vivo/40 specific-pathogen-free male Sprague Dawley rats; 4 weeks	Bone formation was observed	[63]
Lovastatin (1.2 mg/layer)	β-TCP/PCL (β-tricalcium phosphate/polycaprolactone) microchips and PCL nanofiber membranes	In vivo/24 ovariectomized New Zealand rabbits; 12 weeks	Bone parameters significantly improved	[60]
Simvastatin (0.5 μM)	BPPD (bis(PLGA-*phe*-PEG)-qDETA)	In vitro/bone marrow mesenchymal stem cells (BMSCs); 6 days	Promotion of osteogenesis in BMSCs was observed	[96]
Simvastatin (4 mg)	Chitosan	In vitro/BMSC culture; 14 days In vivo/6 healthy ovariectomized rats; 8 weeks	In vitro: A positive effect on cell proliferation was noticed In vivo: The bone regeneration process was observed	[97]
Simvastatin (0.25 mg)	Chitosan	In vivo/21 healthy Sprague Dawley rats; 8 weeks	No significant difference between the control and experimental groups was found	[98]
Simvastatin (0.05 mg)	Chitosan	In vitro/human bone marrow mesenchymal stem cells (hbMMSCs); 14 days	Chitosan scaffold is a bioactive compatible material with regenerative potential for hBMMSCs	[99]
Simvastatin (5 mg/0.5 mL)	Chitosan	In vivo/12 healthy male albino New Zealand rabbits; 6 weeks	The process of bone regeneration was noticed	[100]
Simvastatin (2.5 mg/mL)	Collagen graft	In vivo/9 healthy New Zealand white rabbits; 14 days	An osteoinductive effect was noticed	[101]
Simvastatin (2.5 mg/mL)	Collagen matrix	In vivo/14 healthy New Zealand white rabbits; 14 days	New bone formation was observed	[102]
Rosuvastain (0.1, 0.5, and 2.5 mg/mL)	Collagen sponges	In vivo/18 healthy New Zealand white female rabbits; 4 weeks	Stimulation of bone formation occurred	[64]
Simvastatin (1% concentration)	Gel (composed of polymer 2% HPMC K100M and 20% poloxamer 407)	In vivo/72 healthy Sprague Dawley rats; 56 days	Bone regeneration was observed	[103]
Simvastatin (250 μg)	Gelatin hydrogel	In vivo/60 healthy virgin female Sprague Dawley rats; 8 weeks	Acceleration of fracture healing was observed	[104]
Fluvastatin (1 mM)	Gelatin hydrogel	In vivo/60 healthy male Sprague Dawley rats; 4 weeks	Induced osteogenesis in rat calvarial bone was observed	[67]
Simvastatin (2.5 mg/mL dissolved in 0.2 mL water)	Gelatin hydrogel	In vivo/20 adult male Japanese white rabbits; 12 weeks	New bone formation was observed	[95]
Simvastatin (125 μg)	Gelatin hydrogel	In vivo/42 healthy mature mature Japanese rabbits; 8 weeks	Promotion of tendon–bone healing at an early stage via angiogenesis and osteogenesis occurred but did not affect the biomechanical property in the long term	[105]
Simvastatin (0.5 μM)	GNTS (gelatin-nanofibrillar cellulose- β tricalcium phosphate)	In vivo/30 healthy male Sprague Dawley rats; 8 weeks	Newly formed bone structures were noticed	[106]
Simvastatin (100 nM)	Methylated β-cyclodextrins	In vitro/MC3T3-E1 cells; 14 days	ALP production and the expression of bone sialoprotein and osteocalcin were noticed	[107]
Simvastatin (2.2 mg)	Methylcellulose gel and PLA (polylactide membrane)	In vivo/56 healthy female ICR Swiss mice; 44 days	An increase in bone thickness was observed	[108]
Simvastatin (0.5 mg)	NLC (nanostructured lipid carrier)	In vivo/20 healthy rabbits; 4 weeks	Enhanced bone formation was observed	[109]
Simvastatin (20 μg)	PCL (poly (ε-caprolactone))	In vivo/90 healthy Wistar albino rats; 6 months	An increase in bone mineralization was noticed	[110]
Simvastatin (100 μg/mL)	PCL (poly (ε-caprolactone)) and collagen	In vitro/primary human umbilical vein endothelial cells (pHUVECs); 21 days	Enhanced osteogenic differentiation was noticed	[111]
Simvastatin (2.2 mg)	PCL (poly (ε-caprolactone)) fibrous sheets and structured nanofibers with a gelatin shell	In vivo/24 healthy male New Zealand white rabbits; 12 weeks	Good cell viability and effective osteoinductive and barrier properties were observed	[112]
Simvastatin (5% concentration)	PCL (poly (ε-caprolactone)) nanofibers loaded with polyaniline-coated titanium oxide nanoparticles (TiO_2_/PANI)	In vitro/MC3T3-E1 osteoblast cells; 14 days	Profound cell proliferation was observed	[113]
Simvastatin (dose not stated)	PCL-HA (poly(ε-caprolactone- hydroxyapatite)) microspheres	In vitro/bone marrow mesenchymal stromal cells (BMSCs); 21 days In vivo/3 healthy Sprague Dawley rats; 8 weeks	Osteogenic differentiation of BMSCs was noticed in vitro. Promotion of vascular network and functional bone formation was observed in vivo	[114]
Simvastatin (5% concentration)	PCL-HA (poly(ε-caprolactone-hydroxyapatite)) composite coated on biodegradable Mg alloy nanofibers	In vitro/MC3T3 mouse osteoblast cell line; 7 days	An increase in bone regeneration and control of its degradation occurred	[115]
Simvastatin (from 2.5 × 10^−6^ to 2.5 × 10^−10^ M)	PECL (poly (ethylene glycol))-poly(ε-caprolactone))	In vitro/human osteoblast-like MG-63 cells; 7 days	Osteoblast differentiation and mineralization were observed	[116]
Fluvastatin (0.01 and 0.1 μM)	PEGDM (poly (ethylene glycol) dimethacrylate)	In vitro/human mesenchymal stem cell (hMSC); 14 days	An increase in hMSC CBFA1, ALP, and COL I gene expression was noticed, which indicated an effect on osteogenic differentiation	[68]
Simvastatin (0.5 mg)	PEG-PLA (polyethylene glycol- polylactic acid) polymeric nanomicelles	In vivo/6 healthy New Zealand white rabbits; 4 weeks	Osteoblasts and new capillaries around the trabecular bone were found	[117]
Simvastatin (0.28 and 0.31 μg/mg)	PEG-PLGA (poly (ethylene glycol))-block-poly(lactic-co-glycolic acid)	In vivo/6 healthy ovariectomized Sprague Dawley rats; 12 weeks	A bone formation effect was present	[118]
Simvastatin (2 mg/mL)	PEEK (polyetheretherketone)bio-composite	In vitro/MC3T3-E1 pre-osteoblasts; 14 days	Osteogenic differentiation was observed	[79]
Simvastatin (1 mg/mL)	PET (polyethylene terephthalate)	In vivo/36 healthy New Zealand white rabbits; 8 weeks	Bone healing was observed	[119]
Fluvastatin (75 μg)	PGA (propylene glycol alginate)	In vivo/60 healthy female Wistar rats; 2 weeks	An increase in bone volume was noticed	[69]
Fluvastatin (75 μg)	PGA (propylene glycol alginate)	In vivo/48 healthy female Wistar rats; 4 weeks	An increase in bone–implant contact and mineralized bone volume was observed	[70]
Lovastatin (1 mg/mL)	PGA-PEG (poly(glycolide)-poly(ethylene glycol))	In vitro/mice; 7 days	The study showed that the maximum tolerated dose in mice can be increased	[61]
Simvastatin (5 mg)	PLA (polylactic acid)	In vivo/16 healthy New Zealand white rabbits; 12 weeks	High-density spots were observed and the margins of the defects were more irregular	[120]
Simvastatin (4mg/g PLG)	PLG (poly(lactide-co-glycolide))	In vitro/rat bone marrow cells; 10 days	Bone cell mineralization was observed	[121]
Fluvastatin (0.5 and 1 mg/kg)	PLGA (poly (lactic-co-glycolic acid))	In vivo/40 healthy Sprague Dawley rats; 4 weeks	More bone trabeculae were observed	[71]
Simvastatin (1 mg)	PLGA (poly (lactic-co-glycolic acid))	In vitro/human osteoblastic cell line (hFOB); 11 days	Osteoblastic differentiation was observed	[122]
Simvastatin (0.6% concentration)	PLGA (poly (lactic-co-glycolic acid)) coated around titanium	In vitro/human gingival fibroblasts (HGFs) and stem cells from human exfoliated deciduous teeth (SHEDs); 7 days	High cell viability was observed	[123]
Simvastatin (20 mg/kg)	PLGA (poly (lactic-co-glycolic acid))-encapsulated hydroxyapatite	In vivo/24 healthy female Wistar rats; 45 days	Significant improvement in the bone surface was observed	[124]
Simvastatin (2, 5, and 8% concentrations)	PLGA (poly (lactide-co-glycolide)) microspheres using the electrospraying method	In vitro/human MG-63 osteoblast cells; 7 days	Good biocompatibility of the electrosprayed PLGA microspheres was observed, which increased in the presence of a statin	[125]
Simvastatin (5% concentration)	PLGA (poly (lactic-co-glycolic acid)) microspheres loaded into hydrogel-loaded BCP (biphasic calcium phosphate)	In vitro/MC3T3-E1 pre-osteoblast cells; 7 days	Bone remodeling gene and protein expression were observed	[126]
Simvastatin (3 mg of simvastatin/PLGA)	PLGA (poly (lactic-co-glycolic acid)) with a rapidly absorbable calcium sulfate	In vivo/60 healthy male Sprague Dawley rats; 12 weeks	Osteogenic and angiogenic activity and bone healing process increased	[127]
Simvastatin (0.5 μM)	PLGA-PEG (poly (lactic acid-co-glycolic acid)-polyethylene glycol))	In vitro/BMSCs; 6 days	Improvement in bone healing was observed	[96]
Simvastatin (1 mg)	PLLA (poly-L-lactide)	In vivo/29 healthy male Sprague Dawley rats; 8 weeks	New bone formation and increased bone mineral density were observed	[128]
Simvastatin (~ 120 mg/kg/day)	Polyethylene particles	In vivo/21 healthy female and male C57BL/J6 mice; 14 days	New bone formation was noticed	[129]
Simvastatin (2.2 mg)	Poly(ethylene glycol)-*block*-poly(simvastatin)	In vivo/144 healthy male Sprague Dawley rats; 8 weeks	A significant osteogenic effect was noticed	[130]
Simvastatin (0.5 mM)	Poly (N-isopropylacrylamide) Brush-modified mesoporous hydroxyapatite	In vivo/20 ovariectomized Wistar rats; 6 weeks	Promotion of osteogenesis was observed	[131]
Lovastatin (200 μg/g of foam)	Polyurethane (PUR)	In vivo/6 healthy male Sprague Dawley rats; 4 weeks	An increase in the density of the newly formed bone was observed	[62]
Simvastatin (5 mg/mL)	Polyurethane nanofibers	In vivo/32 healthy adult male Wistar rats; 4 weeks	Induction of bone healing was noticed	[132]
Rosuvastatin (5 mg/mL)	PVA-SF (polyvinyl alcohol–silk fibroin) core-shell nanofibers	In vitro/Human adipose-derived stem cells (hADSCs); 21 days	Improved cell proliferation and osteogenic differentiation occurred	[65]
Simvastatin (2 mg)	SIM-DOME (methylcellulose gel under a polylactic acid dome membrane)	In vivo/44 healthy mature female Sprague Dawley rats; 24 days	New bone formation was observed	[94]
Simvastatin (0.5–1 μM)	Poly(l-lactide-*co*-glycolide)	In vivo/4 healthy male C57/BL/6 J mice; 12 weeks	New bone formation was observed	[133]
Organic non-polymer materials				
Simvastatin (10 mg)	Gelfoam soaked with normal saline	In vivo/50 humans; 12 weeks	An increase in bone density occurred	[134]
Simvastatin (0.1 and 1 mg)	Hyaluronic acid (HA) hydrogels	In vivo/12 healthy male New Zealand rabbits; 8 weeks	A significant influence on osteogenesis was observed	[135]
Simvastatin (0.01, 0.1, and 1 μM)	Injectable tissue-engineered bone (ITB)	In vitro/human adipose-derived stromal cells (hADSCs); 14 days In vivo/26 healthy BALB/C homozygous nude mice; 4 weeks	Osteoblastic differentiation in vitro and bone formation in vivo were observed	[136]
Rosuvastatin (3 mg/mL)	SF (silk fibroin) nanofibers	In vitro/Human adipose-derived stem cells (hADSCs); 21 days	Osteogenic gene differentiation was observed	[66]

## 4. Conclusions

The treatment of large bone defects and effective bone tissue regeneration remain challenges in orthopedics and dental surgery. Constant efforts are made to identify both bone substitute materials with appropriate physicochemical, mechanical, and biological properties, as well as factors that would expedite the process of new bone tissue formation and the development of blood vessels within it.

Therefore, the discovery that statins, drugs that have long been used to lower cholesterol levels in the blood, exhibit strong proangiogenic and osteogenic effects has generated significant interest in their potential use as supportive factors in the treatment of serious bone disorders. Our literature review has demonstrated that statins, acting pleiotropically through various mechanisms, attenuate osteoblast apoptosis, influence the reduction in osteoclast differentiation, and stimulate bone tissue formation, among other effects, by promoting osteoblast differentiation.

Of course, like most drugs, statins exhibit numerous adverse effects, which can be particularly hazardous when high doses are required for the therapy of bone disorders. Hence, much research is focused on developing local delivery systems for statins to bone tissue, which, in addition to minimizing adverse effects, would improve bioavailability and allow for a reduction in drug dosage.

The results of this review suggest that carriers based on inorganic materials (primarily various calcium phosphates and calcium sulfate), polymeric materials (especially chitosan, collagen poly-e-caprolactone, and poly(lactide-co-glycolide)), and inorganic–organic composite materials are capable of delivering statins locally to bone tissue, releasing them over time, and significantly enhancing bone tissue regeneration compared to mere filling. The findings from these studies are promising and highlight the need for further investigation. It is worth noting that only one human study has been published so far; thus, clinical trials should proceed.

While there are numerous works in the literature dedicated to the issue of bone carriers for statins, there are no articles focusing on the combination of different antiresorptive and osteogenic factors with statins. Future research could be dedicated to examining the impact of the simultaneous administration of statins and other factors stimulating bone tissue regeneration (e.g., magnesium ions, strontium, or silicates). These could be introduced in the form of substituted calcium phosphate. Future studies should also focus on determining the appropriate statin dosage under in vivo conditions to ensure therapy effectiveness and safety. It is worth noting that the majority of studies only address one statin, simvastatin. Therefore, further research should also include other representatives of this group.

## Figures and Tables

**Figure 1 ijms-25-02378-f001:**
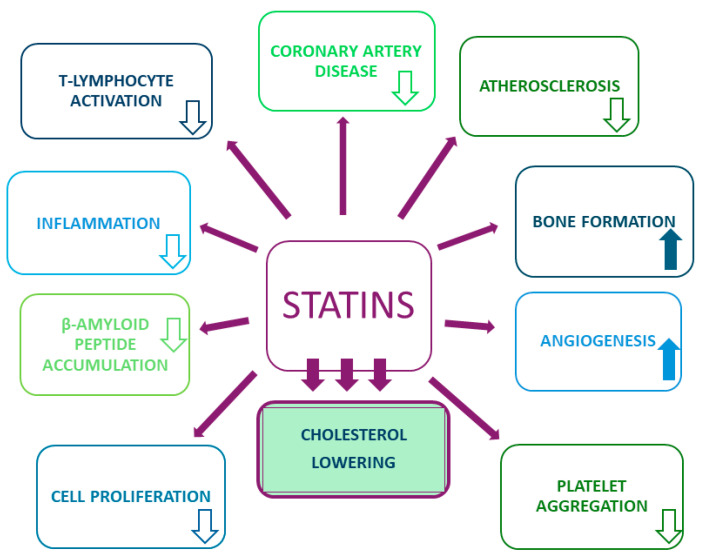
Main and pleiotropic effects of statins (down arrow—decrease, up arrow—increase).

**Figure 2 ijms-25-02378-f002:**
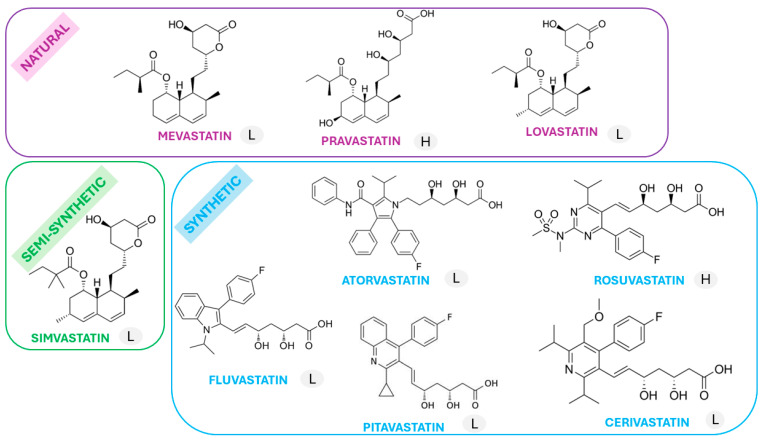
Chemical structure of statins (L—lipophilic statin; H—hydrophilic statin).

**Figure 3 ijms-25-02378-f003:**
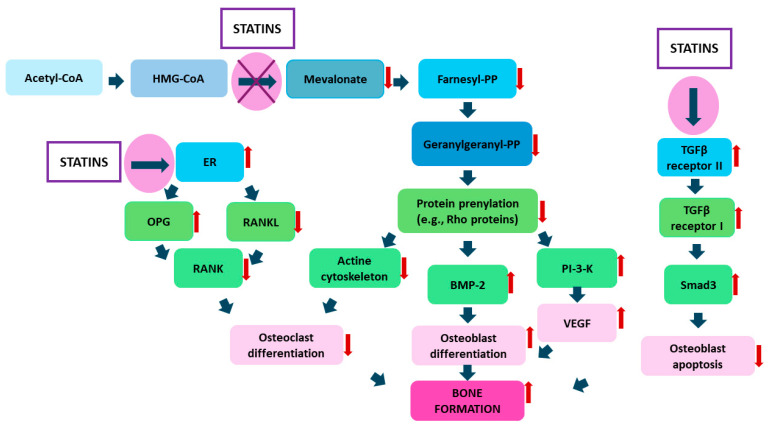
Mechanism of statins’ actions on bone metabolism (Abbreviations: BMP-2—bone morphogenetic protein 2; ER—endoplasmic reticulum; HMG-CoA—3-hydroxy-3-methyl-glutaryl-coenzyme A; OPG—osteoprotegerin; PI-3-K—phosphatidylinositol 3-kinase; PP—pyrophosphate; RANK—receptor activator of NF-κB; RANKL—receptor activator of NF-κB ligand; Smad3—SMAD family member 3; TGFβ—transforming growth factor beta; VEGF—vascular endothelial growth factor). Red down arrow—decrease, red up arrow—increase.

## Data Availability

The data are contained within the article.

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
