# Peer review of "Statins—Their Role in Bone Tissue Metabolism and Local Applications with Different Carriers"

_ijms, 2024, doi:10.3390/ijms25042378_

Round 1

Reviewer 1 Report

Comments and Suggestions for Authors

In the introduction, the authors state that "The paper outlines various carrier types, characterising their structures and underscoring various statins' potential as local treatments for bone diseases." Many readers will expect this to be the case.

In particular, the description in Section 3. Statins Local Delivery Methods and Carriers should be reinforced. The author should present a detailed description of the types and characteristics of carriers and their application methods, with specific details based on the references cited. Alternatively, a detailed description of those mentioned in the references listed in Table 1 that are representative or that the author considers useful or effective would be very useful for many readers. Consideration should also be given to adding explanations with easy-to-understand diagrams and schemas.

Author Response

The authors are grateful for all the remarks and the possibility to response to them. In reply, the following document was prepared. The appropriate changes in the text were made and highlighted with a red colour. Short comments, explanations and responses to the remarks, which were not placed in the main text, were highlighted in blue.

Reviewer 1

In the introduction, the authors state that "The paper outlines various carrier types, characterising their structures and underscoring various statins' potential as local treatments for bone diseases." Many readers will expect this to be the case.

In particular, the description in Section 3. Statins Local Delivery Methods and Carriers should be reinforced. The author should present a detailed description of the types and characteristics of carriers and their application methods, with specific details based on the references cited. Alternatively, a detailed description of those mentioned in the references listed in Table 1 that are representative or that the author considers useful or effective would be very useful for many readers. Consideration should also be given to adding explanations with easy-to-understand diagrams and schemas.

We thank the Reviewer for this comment. We have added the text as followe:

Local administration of statins in the case of bone tissue disorders seems justified for several reasons. Firstly, bone tissue is highly vascularized, and systemic treatment requires the use of large doses of therapeutic substances to achieve an adequate concentration at the target site [54,55]. Therefore, direct administration of statins to bone tissue allows for a reduction in dosage compared to systemic administration [56]. Additionally, the carrier for intraosseous drug delivery can provide a matrix for the infiltration of mesenchymal cells and often serves as a filling material for bone defects [57,58]. It is essential to emphasize that the optimal carrier should exhibit appropriate degradation rates to ensure proper growth of newly formed bone tissue while preventing the formation of fibrous tissue and fibrous encapsulation of the carrier. Local administration of statins can effectively reduce the risk of adverse effects, primarily myopathy characterized by muscle pain and weakness, elevation of creatine kinase (CK) levels in the serum, and, in extreme cases, rhabdomyolysis, which is a life-threatening condition [57,58].

It is worth noting that our study did not focus on the specific doses of statins used in individual research. The results regarding statin dosage in the treatment of bone tissue diseases remain inconclusive. Depending on the dose, these drugs can exhibit both positive and negative effects on bone health. For instance, high doses of simvastatin (20 mg/kg bw/day) have been shown to stimulate new bone tissue formation, while low doses (1 mg/kg bw/day) may inhibit reconstruction and increase bone resorption [59]. Despite promising results, the practical applications of statins in treating bone diseases require further research, including optimizing dosage and exploring effective methods of application.

Table 1 presents the literature review results regarding carriers for statin delivery into the bone tissue. The data indicate that statin delivery systems used mainly simvastatin, in a few cases lovastatin [82,113,128], pitavastatin [81], rosuvastatin [90,130,136], and fluvastatin [93,106,111,112,116].

Carriers were equipped with statins in various ways. The simplest method involves impregnating bone scaffolds, but this has the drawback of rapid drug dissolution in the bloodstream, leading to reduced bioavailability. Alternatively, spray techniques and ultrasonic dispersion can coat statins onto bone scaffolds, allowing for only low-dose application. Other approaches involve micro- and nanospheres, chemical compound combinations, polymerization, and blending with biomaterials, each with its advantages and limitations.

Due to the chemical composition, carriers can be categorized into those made of inorganic materials, natural polymers, synthetic polymers, and polymer/inorganic material composites. The choice of carrier materials is dictated by the structure and properties of bone tissue, composed of an organic matrix (mainly collagen type I fibres) and an inorganic phase, primarily biological apatite, a nanocrystalline calcium phosphate. Inorganic carriers offer high compatibility, bioactivity, and bioresorbability, but may lack sufficient mechanical strength, especially when serving as bone defect fillers. On the other hand, polymers, despite high biocompatibility, may lack the necessary hardness. Therefore, composite carriers seem to be the optimal solution, providing adequate mechanical properties, biodegradation, and controlled statin release at the target site.

In the case of inorganic carriers, calcium phosphates were primarily utilized due to their similarity to biological apatite. Among the most commonly used calcium phosphates, hydroxyapatite and crystalline calcium orthophosphates (in both low- and high-temperature forms, respectively β and α) can be mentioned, with α-TCP being a compound much more soluble than β-TCP and hydroxyapatite. α-TCP has been employed in the form of bone cement, a material that solidifies in reaction with water to form a hard filling for bone defects. β-TCP has also been used in the form of cement as well as a biphasic material with hydroxyapatite, thereby increasing the material's bioresorbability and bioactivity. In the study [71], unconventional hydroxyapatite in the form of nanofibers was used, while in [72], together with simvastatin, it served as a coating for a titanium implant. In this case, hydroxyapatite was used as a factor improving osteointegration and as a carrier for delivering statin. Studies [69] and [70] focus on the use of a scaffold of calcium sulfate loaded with simvastatin. Research conducted using inorganic materials was carried out both in vitro conditions (primarily on BMSCs cells) and in vivo animal models (rats, mice, and rabbits). In all presented studies, simultaneous administration of the inorganic material with statin resulted in increased formation of new bone tissue, increased density of the created tissue, and improvement in bone tissue regeneration, which were observed between the 3rd and 12th week of the experiment.

Significantly more research has focused on the utilization of both natural and synthetic polymers, as well as composite materials, as potential carriers for statins. This emphasis likely stems from the inherent drawbacks of inorganic materials (primarily calcium phosphates), notably their brittleness and thus insufficient mechanical strength, which render them unsuitable for areas subjected to high stresses. Polymers are characterized by high flexibility, whereas calcium phosphate/polymer composites combine the hardness of inorganic materials with the elasticity of polymers, ensuring excellent mechanical properties akin to those of bone tissue, which is also a natural composite.

Among natural polymers utilized for statin carriers, chitosan has been predominant, with collagen or prepared atelocollagen being alternative materials. Gelatin has also been employed in hydrogel form. Concerning synthetic polymers, a wide array has been utilized, including both biodegradable (e.g., PLA, PLLA, PCL, PLGA) and non-biodegradable (PUR, PEEK) polymers. Composite materials described in the available literature consist of hydroxyapatite or beta-TCP combined with polymers such as poly(e-caprolactone), poly(glycerol sebacate), gelatin-nanofibrillar cellulose, PCL, or PLGA. Polymer and composite materials were prepared in various forms, including porous scaffolds, microspheres, nanomicelles, fibers, nanoparticles and coatings on titanium surfaces. Available literature predominantly comprises in vivo studies on rats and rabbits, which consistently indicated increased mineralization, bone tissue formation, and improvement in bone parameters in each case.

A significant aspect investigated under laboratory conditions was the release of statins from previously obtained carriers. Various release profiles were observed depending on the chemical composition of the carrier, its form, and preparation method. Generally, the constructed carriers were designed to release statins gradually over an extended period (from 50-90% within 7 days) [63,137,138]. In the study by [138], a typical microporous structure of calcium phosphate foam was employed, which proved to be a suitable carrier for pitavastatin: the "burst release" effect was relatively minimal, followed by a gradual release of the statin over 72 hours. Various approaches were applied to prolong the release time. For instance, delayed release was achieved in carriers derived from biomimetic beta-TCP by coating it with an additional layer of apatite [63]. This resulted in a matrix that released 20% less simvastatin over 7 days. In the study by [109], a significantly prolonged release of simvastatin was obtained from PCL microspheres contained within a collagen coating covering PET polymer (constructing artificial ligaments). Over 14 days, slightly over 50% of the drug was released, and approximately 75% within nearly 40 days.

Reviewer 2 Report

Comments and Suggestions for Authors

Please refer to my comments and suggestion to improve this manuscript in the attached file.

Comments on the Quality of English Language

Moderate English checks are necessary to ensure the quality of English language. 

Author Response

The authors are grateful for all the remarks and the possibility to response to them. In reply, the following document was prepared. The appropriate changes in the text were made and highlighted with a red colour. Short comments, explanations and responses to the remarks, which were not placed in the main text, were highlighted in blue.

Reviewer 2

  1. For Figure 1, please differentiate between main effect and pleiotropic effect of statins. I would suggest to do; Statins main effect: for cholesterol lowering. Then, please add the pleiotropic effects as in the figure 1.

We thank the Reviewer for this remark. We have changed the Figure 1 according to your suggestion. Please, see the manuscript.

  1. For figure 1, the caption stated there is Statins main biological activities, but in the text, it is mentioned as “The pleiotropic effects … (line 51). The caption could be changed by stating putting as the main and pleiotropic effects of statins.

We are grateful for this suggestion. We have changed the caption as follows:

Figure 1. Main and pleiotropic effects of statins.

  1. Line 57: Please put the reference for the statement “Notably, variations in their chemical structures impact lipophilicity, hydrophilicity, and subsequent absorption, distribution, metabolism, and excretion (XX]. Please also add examples of how the chemical structures could impact the ADME of statins.

We thank the Reviewer for this remark. We have added the following references:

Notably, variations in their chemical structures impact lipophilicity, hydrophilicity, and subsequent absorption, distribution, metabolism, and excretion [14,16]

[16] Climent, E.; Benaiges, D.; Pedro-Botet, J. Hydrophilic or lipophilic statins? Front. Cardiovasc. Med. 2021 8: 687585.Moreover, we have added the text as follows:The molecule of each statin consists of three main parts: the HMG-CoA analogue, a complex ring structure responsible for binding the statin to HMG-CoA reductase, and a side chain structure determining solubility. Statins vary greatly in solubility due to the presence/absence of polar moieties in their predominantly hydrophobic backbones. Lipophilic statins can easily penetrate deeper into cell membranes, enter cells by passive diffusion and are therefore widely distributed in various tissues. They are metabolized by cytochrome P450 (CYP) enzymes upon binding to the membrane. In turn, hydrophilic statins remain bound to the polar surface of the membrane and require transport of proteins into the cell, show greater hepatoselectivity and lower potential for uptake by peripheral cells than lipophilic statins, and are mostly eliminated without modification.

  1. Line 68: Please define “OATP2 transporter”.

We thank the Reviewer for this remark. We have added the text as follows:

Serving as substrates for CYP3A4 (cytochrome P450 family 3 subfamily A member 4) isoenzyme and the OATP-2 (organic anion transporting polypeptide-2), statins can engage in clinically significant interactions with other drugs affecting these proteins.

  1. Line 81: Our study focuses on investigating the effects of simvastatin on bone tissue metabolism considering different carriers for drug application.

In the table of evidence, other statins are also included, not only simvastatin. Better to change “simvastatin” to “statin” in this sentence.

We agree with the Reviewer. We have changed the “simvastatin” to “statin”.

  1. Line 78: Please add evidence why statin needs to be given in carrier. Alternatively, you can relate with the adverse effects of statin.

We thank the Reviewer for this comment. We have added the following text:

It should be noted that statins, like most drugs, are typically administered orally. However, oral administration of statins presents drawbacks, such as first-pass metabolism in the liver and degradation in the gastrointestinal tract, resulting in limited bioavailability. Another concern is the occurrence of side effects such as myopathy, kidney and liver damage, and rhabdomyolysis. Hence, alternative routes of statin administration are under investigation, particularly for potential use in bone diseases where prolonged action directly within the bone tissue is required.

  1. Line 79-89: Please rearrange this whole text into the subtopic 2.0 Statins - role in bone tissue metabolism. I suggest to put this as the first paragraph of this subtopic.

We thank the Reviewer for this suggestion. However, we decided to keep the paragraph in the Introduction section. This text concerns technical data, the purpose of our review and the method of searching literature databases.

  1. Line 187: “It is worth noting that our study did not focus on the specific doses of statins used in individual research.” Please explain why specific doses are not focused in this study? In the abstract, statin dose has been mentioned accordingly. As this study includes in vivo study, alternatively, you can put the method for extrapolation dose from animal to human dose accordingly, where appropriate.

We thank the Reviewer for this comment. We have added the data on specific doses used during the research (please, see Table 1)

  1. Line 214: Please do elaboration text for each study mentioned in the table. The effects of statin combined with different carriers should be discussed accordingly in the text and should not be only stated in the table of evidence.

We thank the Reviewer for this suggestion. We have added the expanded text in Section 3. Please, see the manuscript.

We have added the text as follows:

Table 1 presents the literature review results regarding carriers for statin delivery into the bone tissue. The data indicate that statin delivery systems used mainly simvastatin, in a few cases lovastatin [82,113,128], pitavastatin [81], rosuvastatin [90,130,136], and fluvastatin [93,106,111,112,116].

Carriers were equipped with statins in various ways. The simplest method involves impregnating bone scaffolds, but this has the drawback of rapid drug dissolution in the bloodstream, leading to reduced bioavailability. Alternatively, spray techniques and ultrasonic dispersion can coat statins onto bone scaffolds, allowing for only low-dose application. Other approaches involve micro- and nanospheres, chemical compound combinations, polymerization, and blending with biomaterials, each with its advantages and limitations.

Due to the chemical composition, carriers can be categorized into those made of inorganic materials, natural polymers, synthetic polymers, and polymer/inorganic material composites. The choice of carrier materials is dictated by the structure and properties of bone tissue, composed of an organic matrix (mainly collagen type I fibres) and an inorganic phase, primarily biological apatite, a nanocrystalline calcium phosphate. Inorganic carriers offer high compatibility, bioactivity, and bioresorbability, but may lack sufficient mechanical strength, especially when serving as bone defect fillers. On the other hand, polymers, despite high biocompatibility, may lack the necessary hardness. Therefore, composite carriers seem to be the optimal solution, providing adequate mechanical properties, biodegradation, and controlled statin release at the target site.

In the case of inorganic carriers, calcium phosphates were primarily utilized due to their similarity to biological apatite. Among the most commonly used calcium phosphates, hydroxyapatite and crystalline calcium orthophosphates (in both low- and high-temperature forms, respectively β and α) can be mentioned, with α-TCP being a compound much more soluble than β-TCP and hydroxyapatite. α-TCP has been employed in the form of bone cement, a material that solidifies in reaction with water to form a hard filling for bone defects. β-TCP has also been used in the form of cement as well as a biphasic material with hydroxyapatite, thereby increasing the material's bioresorbability and bioactivity. In the study [71], unconventional hydroxyapatite in the form of nanofibers was used, while in [72], together with simvastatin, it served as a coating for a titanium implant. In this case, hydroxyapatite was used as a factor improving osteointegration and as a carrier for delivering statin. Studies [69] and [70] focus on the use of a scaffold of calcium sulfate loaded with simvastatin. Research conducted using inorganic materials was carried out both in vitro conditions (primarily on BMSCs cells) and in vivo animal models (rats, mice, and rabbits). In all presented studies, simultaneous administration of the inorganic material with statin resulted in increased formation of new bone tissue, increased density of the created tissue, and improvement in bone tissue regeneration, which were observed between the 3rd and 12th week of the experiment.

Significantly more research has focused on the utilization of both natural and synthetic polymers, as well as composite materials, as potential carriers for statins. This emphasis likely stems from the inherent drawbacks of inorganic materials (primarily calcium phosphates), notably their brittleness and thus insufficient mechanical strength, which render them unsuitable for areas subjected to high stresses. Polymers are characterized by high flexibility, whereas calcium phosphate/polymer composites combine the hardness of inorganic materials with the elasticity of polymers, ensuring excellent mechanical properties akin to those of bone tissue, which is also a natural composite.

Among natural polymers utilized for statin carriers, chitosan has been predominant, with collagen or prepared atelocollagen being alternative materials. Gelatin has also been employed in hydrogel form. Concerning synthetic polymers, a wide array has been utilized, including both biodegradable (e.g., PLA, PLLA, PCL, PLGA) and non-biodegradable (PUR, PEEK) polymers. Composite materials described in the available literature consist of hydroxyapatite or beta-TCP combined with polymers such as poly(e-caprolactone), poly(glycerol sebacate), gelatin-nanofibrillar cellulose, PCL, or PLGA. Polymer and composite materials were prepared in various forms, including porous scaffolds, microspheres, nanomicelles, fibers, nanoparticles and coatings on titanium surfaces. Available literature predominantly comprises in vivo studies on rats and rabbits, which consistently indicated increased mineralization, bone tissue formation, and improvement in bone parameters in each case.

A significant aspect investigated under laboratory conditions was the release of statins from previously obtained carriers. Various release profiles were observed depending on the chemical composition of the carrier, its form, and preparation method. Generally, the constructed carriers were designed to release statins gradually over an extended period (from 50-90% within 7 days) [63,137,138]. In the study by [138], a typical microporous structure of calcium phosphate foam was employed, which proved to be a suitable carrier for pitavastatin: the "burst release" effect was relatively minimal, followed by a gradual release of the statin over 72 hours. Various approaches were applied to prolong the release time. For instance, delayed release was achieved in carriers derived from biomimetic beta-TCP by coating it with an additional layer of apatite [63]. This resulted in a matrix that released 20% less simvastatin over 7 days. In the study by [109], a significantly prolonged release of simvastatin was obtained from PCL microspheres contained within a collagen coating covering PET polymer (constructing artificial ligaments). Over 14 days, slightly over 50% of the drug was released, and approximately 75% within nearly 40 days.

 Please make sure to put in title for the table. eg: Table 1. The effects of various statins combined with different types of carriers on bone.

The title for Table 1 was added: The effects of various statins combined with different types of carriers on bone.

  1. In the table, for the reference no 65, it was stated that it is a meta-analysis study. Please include only research articles. Please remove this study accordingly. However, if the outcome of the meta-analysis is important, please include it as supporting evidence.

We agree with the Reviewer that only research articles should be included, so reference no. 65 was deleted.

  1. In the table, in terms of model of study, please provide the species of the rats. And if available, please also state the condition of animal/cell culture used. Example: ovariectomised Sprague Dawley rats. If no specific condition, please mention it as healthy Sprague Dawley rats. Please be more specific to do the table of evidence.

We are grateful for this comment. According to the suggestion, the doses of statins were added. Species of animals and their condition were specified if the source article provided this information.

  1. In the table, please specify the dose for the statins used in each study mentioned.

We thank the Reviewer for this remark. The doses/concentrations of statins were added according to information provided by the source article.

  1. please rephrase the “comment” column. I would suggest to label the column as “Findings”

We thank the Reviewer for this suggestion. The “Comment” column was renamed as “Findings”.

  1. For the table, I would suggest to put the duration of treatment in the same column of model of study. Model of study/ Duration of treatment. Therefore, in the “findings” column for each study, only the finding/outcome should be stated.

We are grateful for this comment. “Duration of treatment” was added to the column of “Model of study” as suggested so now “Findings” column presents only results of studies.

  1. Conclusions: Please do a brief conclusion. In a conclusion, there should not be statement with the cited reference like this.

“According to literature reports, a crucial factor in the action of statins on bone tissue is their dosage. Studies have indicated that the optimal concentration of simvastatin for promoting osteogenic differentiation of hADSCs (human adipose-derived stromal cells) under in vitro conditions is 1 μM (also the threshold cytotoxic concentration).” I would suggest to rearrange these sentences in the text accordingly, which could be supporting evidence for any study stated.

We agree with the Reviewer that this paragraph is unnecessary. We have deleted it.

  1. Line 255: In conclusion part, please comment whether there are any clinical studies performed using statin combined with carrier for the bone effects.

We have added the following sentence:

It is worth noting that only one human study has been published so far, thus clinical trials should proceed.
